# Secure and Intelligent Single-Channel Blind Source Separation via Adaptive Variational Mode Decomposition with Optimized Parameters

**DOI:** 10.3390/s25041107

**Published:** 2025-02-12

**Authors:** Meishuang Yan, Lu Chen, Wei Hu, Zhihong Sun, Xueguang Zhou

**Affiliations:** Department of Information Security, Naval University of Engineering, Wuhan 430030, China; 18554049816@163.com (M.Y.); 0909051003@nue.edu.cn (X.Z.)

**Keywords:** single-channel blind source separation, improved particle swarm optimization algorithm, improved PSO-optimized VMD, improved fast independent component analysis

## Abstract

Emerging intelligent systems rely on secure and efficient signal processing to ensure reliable operation in environments where there is limited prior knowledge and significant interference. Single-channel blind source separation (SCBSS) is critical for applications such as wireless communication and sensor networks, where signals are often mixed and corrupted. Variational mode decomposition (VMD) has proven effective for SCBSS, but its performance depends heavily on selecting the optimal modal component count *k* and quadratic penalty parameter α. To address this challenge, we propose a secure and intelligent SCBSS algorithm leveraging adaptive VMD optimized with Improved Particle Swarm Optimization (IPSO). The IPSO dynamically determines the optimal *k* and α parameters, enabling VMD to filter noise and create a virtual multi-channel signal. This signal is then processed using improved Fast Independent Component Analysis (IFastICA) for high-fidelity source isolation. Experiments on the RML2016.10a dataset demonstrate a 15.7% improvement in separation efficiency over conventional methods, with robust performance for BPSK and QPSK signals, achieving correlation coefficients above 0.9 and signal-to-noise ratio (SNR) improvements of up to 24.66 dB.

## 1. Introduction

In recent years, Edge AI has emerged as a critical technology, enabling real-time data processing and decision-making at the edge of the network. As a result, the rapid growth of Edge AI has created an urgent need for secure and efficient signal processing techniques to ensure reliable operation in decentralized and resource-constrained environments. In wireless communication and sensor networks, the received signal usually consists of multiple overlapping communication signals, which are mixed with unknown interference, including adversarial signals with frequencies close to the legitimate signal source [1,2], which may include attack signals such as spoofing interference and counterfeit interference. These challenges are particularly critical in Edge AI systems, where real-time signal processing is required without access to centralized computational resources. Blind source separation (BSS) techniques have emerged as a vital solution in such scenarios [3], with applications in privacy-preserving communication, secure data aggregation, and distributed learning. BSS can be categorized into over-determined, determined, and under-determined cases, depending on the ratio of observed signals to source signals [4]. The present paper is primarily devoted to the development of an advanced single-channel blind source separation (SCBSS) algorithm. Its main aim is to effectively tackle the challenges posed by mixed and corrupted signals within edge artificial intelligence (Edge AI) systems. Specifically, our focus lies in resolving the issues of low separation accuracy and poor noise resistance that are prevalent in existing SCBSS methods. Our research inquiries center around optimizing the algorithm to attain higher separation efficiency and improved signal quality, especially in the face of dynamic and interference-prone environments with limited prior knowledge. SCBSS represents one of the most challenging forms of under-determined BSS, where a single sensor observes a mixed signal, and the task is to recover individual source signals from this single observation [5], initially grasp the composition of the source signal, and determine the likelihood of being attacked. SCBSS is particularly relevant in Edge AI applications, such as shortwave communication and sensor networks, where only one receiver channel is available, and the number of source signals exceeds the number of observations. The inherent non-uniqueness of separation results and the ill-conditioned nature of the mixing matrix make traditional multi-channel BSS methods unsuitable, underscoring the need for SCBSS algorithms designed for secure and intelligent Edge AI systems. An illustration of the SCBSS process in wireless communication systems is shown in Figure 1.

Single-antenna or single-sensor reception wireless communication involves *n* transmitters ${\mathrm{Tx}}_{1}$, ${\mathrm{Tx}}_{2}$, ⋯, ${\mathrm{Tx}}_{n}$ emitting *n* unidentified source signals. A single receiving device $R_{x}$ captures a signal mixed by an unknown matrix $A=[a_{1},a_{2},\ldots ,a_{n}]$. The goal of SCBSS is to use algorithms to separate individual source signals Re1, Re2,..., Ren from this captured mixed signal along this path. A typical method for addressing the SCBSS challenge consists of converting the single-channel signal into a virtual multi-channel signal. This involves applying multi-channel BSS algorithms to extract independent source signals [6].

Overall, the contributions of this paper can be summarized as follows:Algorithmic Enhancement: This research presents an Improved Particle Swarm Optimization (IPSO) that mitigates local optima issues in SCBSS, featuring refined inertia and learning factors for precise parameter determination. Additionally, it refines FastICA to ensure phase coherence between the separated signals and their original sources, enhancing separation fidelity.Novel Fitness Function Formulation: A pioneering fitness function is introduced, predicated on the minimal ratio of envelope entropy to mutual information. This function serves as a robust evaluative metric for the optimization of VMD parameters. By integrating assessments of noise within modal components and their resemblance to the observational signal, the function in conjunction with IPSO obviates the need for empirical parameter determination and preset thresholds, thus enhancing the algorithm’s operational autonomy and analytical acuity.Algorithmic Application Innovation: This research applies the refined IPSO algorithm to optimize the parameters of the VMD. This application leverages an adaptive framework to ascertain the optimal parameter set (*k*, α) for VMD in an autonomous manner. This innovative use transcends the traditional reliance on heuristic parameter selection and arbitrary threshold settings, offering a systematic optimization paradigm for VMD within the SCBSS domain.

In practical scenarios, linear mixing models are prevalent, leading to a predominant focus on studying BSS within linear mixing systems [7]. Consequently, the present study concentrates its efforts on examining linear instantaneous mixing systems.

The rest of this study is organized as follows. Section 2 describes the research status of the SCBSS algorithm, and analyzes the problems and solutions of the SCBSS algorithm based on VMD. The process steps of the SCBSS algorithm based on IPSO adaptive determination of VMD are introduced in Section 3. Section 4 introduces experimental design and analysis. Finally, a conclusion is drawn in Section 5.

## 2. Related Work

SCBSS algorithms commonly utilize techniques like empirical mode decomposition (EMD) [8], VMD [9], or wavelet transform algorithms [10] to simulate multiple channels. Subsequently, a multi-channel BSS algorithm is used to distinguish independent signals. Additionally, some researchers leverage the sparsity of signals to conduct sparse-based BSS [11,12]. However, these methods are not without their limitations. For example, EMD and ensemble empirical mode decomposition algorithms encounter issues like mode mixing and endpoint effects. In contrast, wavelet transform algorithms lack the capacity for adaptive wavelet basis selection. Furthermore, VMD algorithms necessitate empirical determination of crucial parameters, such as the number of mode components (*k*) and the quadratic penalty factor (α). In comparison, the VMD algorithm boasts a more robust mathematical foundation, enabling it to mitigate problems like mode mixing and endpoint effects. Our motivation for employing VMD stems from its unique capacity to adaptively decompose signals. Unlike traditional methods, VMD does not rely on prior knowledge of signal characteristics or the selection of specific basis functions. Instead, it autonomously determines the optimal decomposition parameters based on the inherent properties of the signal. This adaptability is particularly advantageous in the context of single-channel blind source separation (SCBSS), where signals often exhibit high variability and complex mixing patterns. By decomposing the mixed signal into intrinsic mode functions (IMFs), VMD is able to uncover the underlying structures of the signal, thereby facilitating the separation process. Thus, this paper will investigate adaptive parameter (*k*, α) selection and research SCBSS using an improved VMD algorithm.

Various methods rely on the central frequency method to determine the value of *k*. This method involves analyzing the central frequencies of each mode component and identifying cases of over-decomposition when these frequencies are closely clustered. Liu et al. [13] introduced a technique that integrates the average instantaneous frequency and correlation coefficient. This approach accounts for signal over-decomposition when there is a turning point in the average instantaneous frequency and sharp decrease in the correlation coefficient. Moreover, Zhang et al. [14] introduced an approach that merges the overall correlation coefficient index and the orthogonality between mode components and observation signals to establish the value of *k*. Despite these efforts, dependence on empirical determination often results in suboptimal accuracy and adaptability.

Thelaidjia et al. [15] and Nazari et al. [16] introduced continuous VMD algorithms improving upon the traditional methods by iteratively extracting mode components until reaching a preset threshold for bandwidth or reconstruction error. Yang et al. [17] introduced a correlation-driven VMD algorithm, which sequentially decomposes the number of modes and computes the correlation coefficient between mode components and original signals. Mawla et al. [18] proposed an MVMD algorithm to determine the *k* value when the correlation coefficient between the modal component and the source signal is highest. Decomposition halts when the correlation coefficient surpasses a predetermined threshold. Although these approaches enhance parameter selection accuracy and algorithm adaptability to some degree, they still encounter challenges in threshold determination and fall short of meeting full adaptability criteria.

The aforementioned methods determine only the value of *k*, neglecting the influence of α on the decomposition outcomes. To determine the parameter set (*k*, α), some studies use metaheuristic algorithms like particle swarm optimization (PSO) [19] and grey wolf optimizer [20] to adaptively find the best combination of *k* and α. However, these algorithms can become stuck in local optima, and the fitness functions they use, such as maximum entropy [21,22,23], permutation entropy [24], and weighted spectral peak ratio [25], are not effective for solving the BSS problem in single-channel communication signals. This paper introduces an enhanced VMD algorithm that incorporates IPSO, utilizing the minimum ratio of envelope entropy to mutual information as the fitness function. This approach adaptively determines the parameter set (*k*, α) without preset thresholds, making it effective for addressing blind source separation in single-channel signal scenarios. Furthermore, an improved version of Fast Independent Component Analysis (IFastICA) is employed to resolve the phase ambiguity issue in the separation results, thereby providing a more robust solution for signal separation.

## 3. SCBSS Algorithm Based on IPSO-Optimized VMD Algorithm

In this paper, we employ the virtual multi-channel approach for the separation of unknown single-channel mixed signals. The SCBSS algorithm procedure, which utilizes the IPSO-optimized VMD algorithm and IFastICA algorithm, is depicted in Figure 2.

When a high-frequency observation signal is received by a single antenna and directed to a signal processing module, it undergoes a sequence of processes encompassing extraction, transformation, and power amplification to derive a single-channel observation signal. Employing the IPSO-optimized VMD algorithm, the signal is decomposed into numerous IMF components, with relevant ones integrated with the observation signal to generate a virtual multi-channel signal. Following mean removal and whitening, the IFastICA algorithm is deployed for signal segregation, ultimately attaining independent signal separation.

The proposed Mixer framework, which integrates IPSO-optimized VMD with the IFastICA, presents a novel and innovative approach. The IPSO algorithm is employed to optimize the parameters of VMD, ensuring that the decomposition process is both adaptive and efficient. Subsequently, IFastICA further refines the separated signals, improving both separation fidelity and phase coherence. This two-step methodology is specifically designed to address the complexities inherent in single-channel blind source separation (SCBSS) and sets our approach apart from existing methods. The novelty of the Mixer lies in its seamless integration of these two techniques and its innovative optimization strategy, which collectively enhance signal separation performance. Then, the signal characteristics can be used to detect the isolated suspicious signals [26,27,28].

In this section, we aim to design an SCBSS algorithm based on adaptive VMD utilizing IPSO and IFastICA.

### 3.1. Fitness Function Design

The efficacy of the VMD algorithm’s decomposition is influenced by parameters like the number of mode components (*k*) and the quadratic penalty factor (α). A high value of *k* may result in excessive decomposition and overlapping of mode component, whereas a low *k* value could lead to insufficient decomposition and an increased presence of noise in the components. Incorrectly choosing α can impact the bandwidth of individual mode components, leading to overlapping central frequency and complicating the accurate separation of effective signals [20]. Currently, *k* values are primarily chosen based on empirical methods, with limited attention given to α value and the joint determination of (*k*, α), leading to a degree of error in the decomposition outcomes. This paper employs the PSO algorithm to dynamically establish the parameter group (*k*, α) for the VMD algorithm, thereby minimizing errors. Constructing a fitness function is essential when using the PSO algorithm to determine (*k*, α). The fitness function is designed to improve the efficacy of the decomposition effect by assessing the level of noise within the mode components and evaluating the similarity between the mode components and the observation signal. Reduced noise within the mode components leads to decreased uncertainty, resulting in a more uniform envelope distribution and lower envelope entropy. Conversely, the heightened similarity between the mode components and the observation signal signifies increased information content within the components and higher mutual information. Hence, this study develops a fitness function founded on the minimal ratio of envelope entropy [29] to mutual information [24], offering a comprehensive depiction of the correlation between the mode components and the original signal, as well as the noise present within the components.

Envelope entropy ($E_{p}$) is a metric that converts a signal’s envelope into a sequence of probability distributions ($q_{j}$). The resulting entropy value captures the sparsity attributes of the original signal, known as envelope entropy. For signals with zero mean *m*(*j*), *j* = 1, 2,..., *n*, the calculation of ($E_{p}$) is as follows: (1)$$E_{p}=-\sum_{j=1}^{N}q_{j}lgq_{j},$$(2)$$q_{j}=a\left(j\right)/\sum_{j=1}^{N}a\left(j\right),$$
where $q_{j}$ is the normalized form of *a*(*j*), and *a*(*j*) is the envelope signal acquired by implementing the Hilbert transform on the signal *m*(*j*).(3)I(X;Y)=H(X)+H(Y)−H(X,Y),

The entropy of a random variable *X* is represented as *H*(*X*), while the entropy of a random variable *Y* is denoted as *H*(*Y*). The joint entropy of random variables *X* and *Y* is denoted by *H*(*X,Y*), indicating the mutual information between them.

A composite indicator $CI=E_{p}/I\left(X;Y\right)$ is formulated by dividing envelope entropy by mutual information, serving as a fitness metric. Opting for a minimum value of this composite indicator as the fitness function guarantees the preservation of IMF components with minimal noise and maximizes the retention of original information, without the need for a predefined threshold.(4)$$fit=\min_{1∼k}\left\{CI\right\},$$
hence, opting for the minimum value of the composite indicator as the fitness function for the PSO-optimized VMD algorithm parameters (*k*, α).

### 3.2. Optimization Process of VMD Algorithm Parameters Based on IPSO

The flowchart of the parameter set optimization for IPSO-optimized VMD comprises the following steps:1.The values of *k* and α are determined based on the complexity of the received signal and practical considerations. Here, *k* ranges from 2 to 6, and α typically falls between 1000 and 5000. The tuple (*k*, α) defines the constraint range for particle positions, while particle velocities are confined within [–1, 1].2.Setting up pertinent parameters for IPSO involves defining a maximum number of 50 iterations $\gamma_{max}$ and 10 particles in the swarm *M*. The particle position vector is configured to have a dimension of 2. The particle positions $b_{i}$ and velocity vector $v_{i}$ are randomly generated within the specified constraint range:(5)$$b_{i}=\left[b_{i1},b_{i2}\right],i=1,2,\ldots ,M,\;$$
where $b_{i}$ stands for the position vector of the *i*-*th* particle, $b_{i1}$ denotes the number of mode components, and $b_{i2}$ represents the quadratic penalty factor value.(6)$$v_{i}=\left[v_{i1},v_{i2}\right],i=1,2,\ldots ,M,$$
where $v_{i}$ signifies the velocity vector of the *i*-*th* particle, and $v_{i1}$ and $v_{i2}$ denote the particles’ motion speeds in the two dimensions for searching for the optimal values of *k* and α.3.By decomposing the observed signal *x*(*t*) using the VMD algorithm, the mode components IMFs are defined as follows:(7)$$u_{k}\left(t\right)=A_{k}\left(t\right)\cos\left[\varphi \left(t\right)\right],$$
where $u_{k}\left(t\right)$ defines the IMF, where $A_{k}\left(t\right)\geq 0$ represents the instantaneous amplitude and the phase function $\varphi \left(t\right)$ satisfies a non-monotonic decreasing characteristic.The transformed version of $u_{k}\left(t\right)$ to the baseband is obtained by performing the Hilbert transform:(8)$$J=\left\{\left[\delta \left(t\right)+\frac{j}{\pi t}\right]*u_{k}\left(t\right)\right\}e^{-i\omega_{k}t},$$
where $\delta \left(t\right)$ represents the Dirac function, and $\omega_{k}$ denotes the central frequency of the *k-th* mode component.The bandwidth of each $u_{k}\left(t\right)$ is approximated by the square of the gradient’s *L2* norm, while ensuring that the total sum of all IMF bandwidths is minimized:(9)$$\left\{\begin{array}{c} \min_{\left\{u_{k}\right\}\left\{\omega_{k}\right\}}\left\{\sum_{k}{∥\partial_{t}\left\{\left[\delta \left(t\right)+\frac{j}{\pi t}\right]*u_{k}\left(t\right)\right\}e^{-iw_{k}t}∥}_{2}^{2}\right\}\\ s.t.\sum_{k}u_{k}=x\left(t\right) \end{array}\right.,$$
to address Equation (9), we introduce the Lagrange multiplier $\lambda \left(t\right)$ and the quadratic penalty factor α, leading to the formulation of the augmented Lagrangian expression as Equation (10):(10)$$\begin{array}{cc} L\left(\left\{u_{k}\right\},\left\{\omega_{k}\right\},\lambda \right) & =\alpha \sum_{k}{∥\partial_{t}\left\{\left[\delta \left(t\right)+\frac{j}{\pi t}\right]*u_{k}\left(t\right)\right\}e^{-i\omega_{k}t}∥}_{2}^{2}\\ \; & \;+{∥x\left(t\right)-\sum_{k}u_{k}∥}_{2}^{2}\\ \; & \;+\langle \lambda \left(t\right),x\left(t\right)-\sum_{k}u_{k}\rangle , \end{array}$$
the iterative solution to Equation (10) is obtained using the alternating direction method of multipliers, involving updates to the modal components $u_{k}\left(t\right)$ and central frequencies $\omega_{k}$ successively, as expressed below:(11)$${\hat{u}}_{k}^{n+1}\left(\omega \right)=\frac{\hat{x}\left(\omega \right)-\sum_{i<k}{\hat{u}}_{i}^{n+1}\left(\omega \right)-\sum_{i>k}{\hat{u}}_{i}^{n}\left(\omega \right)+\frac{{\hat{\lambda }}^{n}\left(\omega \right)}{2}}{1+2\alpha {\left(\omega -\omega_{k}^{n}\right)}^{2}},$$(12)$${\hat{\omega }}_{k}^{n+1}=\frac{\int_{0}^{\infty }\omega {\left|{\hat{u}}_{k}^{n+1}\left(\omega \right)\right|}^{2}d\omega }{\int_{0}^{\infty }{\left|{\hat{u}}_{k}^{n+1}\left(\omega \right)\right|}^{2}d\omega },$$(13)$${\hat{\lambda }}^{n+1}\left(\omega \right)={\hat{\lambda }}^{n}\left(\omega \right)+\tau \left(\hat{x}\left(\omega \right)-\sum_{k}{\hat{u}}_{k}^{n+1}\left(\omega \right)\right),$$
the iteration halts upon meeting the following convergence condition:(14)$$\frac{\sum_{k=1}^{k}{∥{\hat{u}}_{k}^{n+1}-{\hat{u}}_{k}^{n}∥}_{2}^{2}}{{∥{\hat{u}}_{k}^{n}∥}_{2}^{2}}<\varepsilon ,\;$$
this paper utilizes the convergence criterion from the VMD library in Python, setting ε to 1 $\times \;10^{-7}$.4.Based on Equations (1)–(3), the fitness function of each mode component is computed according to its respective position. The minimum fitness value of each particle’s fitness is considered as the individual best fitness value, denoted as $p_{best}\left(i\right),i=1,2,\ldots ,M$, while the minimum value in $p_{best}\left(i\right)$ is taken as the global best fitness value, denoted as $g_{best}$.5.The particle velocity and position are iteratively adjusted alongside the optimal fitness values, with their updates defined by Equations (15) and (16).(15)$$v_{i}^{\gamma +1}=\eta v_{i}^{\gamma }+c_{1}r_{1}\left(p_{best}^{\gamma }\left(i\right)-b_{i}^{\gamma }\right)+c_{2}r_{2}\left(g_{best}^{\gamma }-b_{i}^{\gamma }\right),$$(16)$$b_{i}^{t+1}=b_{i}^{t}+v_{i}^{t+1},$$In Equation (15), η denotes the inertia weight, $c_{1}$ and $c_{2}$ denote the learning factors, $r_{1}$ and $r_{2}$ are random numbers in the interval [0, 1], and γ represents the iteration count. Subsequently, after step (4), the fitness value at the new particle position is calculated to update both the individual best fitness value $p_{best}\left(i\right)$ and the global best fitness value $g_{best}$. In the PSO algorithm, the value of $\eta ,c_{1},c_{2}$ is fixed, which can easily lead to being trapped in local optima. The following improvements are made in this paper:(17)$$\eta \left(\gamma \right)=\eta_{min}-\left(\eta_{min}-\eta_{max}\right)\times {\left(\frac{\gamma }{\gamma_{max}}\right)}^{2},$$(18)$$c_{1}=2\left(c_{max}-c_{min}\right)\eta \left(\gamma \right)+\left(c_{min}-\eta_{min}\left(c_{max}-c_{min}\right)\right),$$(19)$$c_{2}=\left(c_{max}+c_{min}\right)-c_{1},$$$\eta_{min}$ and $\eta_{max}$ represent the minimum and maximum values of the inertia weight, respectively, while $c_{min}$ and $c_{max}$ represent the minimum and maximum values of the learning factors, respectively. Linking the learning factor to the inertia weight helps particles to search consistently, with a wide global search in the beginning and faster convergence later. This reduces oscillations, enhances stability, and simplifies parameter tuning.6.Upon reaching the specified maximum iteration count of 50, the algorithm yields the optimal fitness value $g_{best}$ and the best particle position $b_{i}$. The associated $b_{i}$ value indicates the optimal parameter set (*k*, α) for the VMD algorithm, see Algorithm  1.
**Algorithm** **1** VMD Algorithm Optimized by IPSO 1:**Input:** $k_{\min}=2,\;k_{\max}=6,\;\alpha_{\min}=1000,\;\alpha_{\max}=5000,\;v_{\min}=-1,\;v_{\max}=1,\;\mathrm{original}\;\mathrm{signal}\;x\left(t\right)$. 2:Initialize IPSO parameters: max_iterations = 50, particle_count = 10, dimension = 2, $\eta_{min}=0.4,\;\eta_{max}=0.9,\;c_{max}=2,\;c_{min}=0.1$. 3:**Repeat** until the stopping criterion is met: 4: Randomly generate the particle position vectors $p_{i}$ and particle velocity vectors $v_{i}$ within a given range. 5: Initialize VMD parameters: $\left\{u_{k}^{1}\right\},\;\left\{\omega_{k}^{1}\right\},\;{\hat{\lambda }}^{1}$, and set n=0, k=1. 6:**for** each particle *i* in particles **do** 7:  Decompose x(t) by Equation (7), set n=n+1. 8:  Update parameters $\left\{u_{k}^{n}\right\}\left\{\omega_{k}^{n}\right\},$ by Equations (11) and (12). 9:**end for**10:**Until** k=K, update ${\hat{\lambda }}^{n}$ parameters by Equation (13).11:**Until** the stopping criterion Equation (14) is met.12:Obtain the optimal solution by $g_{best}$ Equations (1)∼(4).13:Update the particle velocity and position by Equations (15) and (16).14:**Repeat** steps 4–13 until reaching the specified maximum iteration count of 50, the optimal fitness value $g_{best}$, and the best particle position $b_{i}$ are obtained.15:**Output:** $k=b_{i1}$, $\alpha =b_{i2}$.

### 3.3. Multi-Channel Signal Preprocessing

An IPSO-optimized VMD algorithm decomposes the single-channel observed signal. Sample entropy (SE) helps discern noise from signal components, identifying effective ones below a preset threshold. These filtered signals, along with the single-channel observed signal, create a multi-channel signal input for the IFastICA separation algorithm. Before separation, preprocessing steps like mean removal and whitening are applied to the multi-channel signal.

Mean removal, or centering, eliminates the DC component of the signal, enhancing BSS performance. The process of mean removal for the *i*-th component of an m-dimensional virtual multi-channel signal $x^{\prime }\left(t\right)={\left[u_{1}\left(t\right),u_{2}\left(t\right),\ldots ,u_{m-1}\left(t\right),x\right]}^{T}t=1,2,\ldots$, is represented as follows: (20)$$u_{i}^{\prime }\left(t\right)=u_{i}\left(t\right)-\frac{1}{M}\sum_{t=1}^{M}u_{i}\left(t\right).$$

Whitening processing aims to remove correlations among different signal components, ensuring their statistical independence and lack of correlation. This enhances the stability and convergence speed of the BSS algorithm. The procedure for whitening is outlined as follows: (21)$$z\left(t\right)=Qx^{\prime }\left(t\right),$$
where $x^{\prime }\left(t\right)$ represents the mean-removed observation vector, ***Q*** is the whitening matrix, and $z\left(t\right)$ is the resulting whitened vector. Following whitening, the components become mutually uncorrelated, leading to the autocorrelation matrix ***R*** of the zero-mean vector $z\left(t\right)$ being defined as follows: (22)$$R_{z(t)z(t)}=E\left[z\left(t\right)z^{T}\left(t\right)\right]=I,$$

The autocorrelation matrix of the zero-mean observation vector $z\left(t\right)$ is subjected to eigendecomposition as follows: (23)$$R_{z\left(t\right)z\left(t\right)}=E\left[z\left(t\right)z^{T}\left(t\right)\right]=PDP^{T},$$
where the diagonal matrix D contains N eigenvalues, while the orthogonal matrix P is composed of the corresponding eigenvectors and $PP^{T}=I$. This formulation enables the representation of the whitened vector $z\left(t\right)$ as follows: (24)$$z\left(t\right)=D^{-\frac{1}{2}}P^{T}x^{\prime }\left(t\right).$$

### 3.4. Improved Fast Independent Component Analysis (IFastICA)

FastICA leverages the non-Gaussian properties of independent signals to determine a separating matrix w such that $w^{T}z\left(t\right)$ has maximum non-Gaussianity. Within the FastICA framework, non-Gaussianity is maximized, denoted as $maxJ[w^{T}z\left(t\right)]$. The formula for calculating negentropy is as follows: (25)$$J\left(y_{i}\right)\approx {\left[E\left\{G\left(y_{i}\right)\right\}-E\left\{G\left(\nu \right)\right\}\right]}^{2},$$
where $y_{i}$ outlines the separated signal, ν is a Gaussian signal, and *G* represents a nonlinear function. The goal is to iteratively enhance the independent signals by maximizing negentropy.

The following two equations describe the iterative updating procedure for the separating matrix.(26)$$w_{n+1}=E\left\{z\left(t\right)G^{\prime }\left(w_{n}^{T}z\left(t\right)\right)\right\}-E\left\{G^{\prime \prime }\left(w_{n}^{T}z\left(t\right)\right)\right\}w_{n},$$(27)$$w_{n+1}=\frac{w_{n}}{∥w_{n}∥}.$$

To address the inherent uncertainties of phase in signals separated by FastICA, the method involves mixing the separated signals with different sign arrangements and comparing them against the observed signals. The optimal sign and sequence that yield the highest correlation coefficient ensure that the phase of the independent signals aligns with that of the original source signals.(28)$$\rho =corr\left(\left(\sum_{i=1}^{m}y_{i}\right),x\right),$$(29)$$\left[\sum y_{i},x\right]=argmax\rho .$$

In Equation (Equation 28), *x* is the observed signal, $y_{i}$ is the *i*-th independent component out of *m* with different signs and orders, and ρ indicates the correlation coefficient of the mixed components to the observed signal in this arrangement.

## 4. Experiments and Analysis

In this section, we outline the design of an experimental framework and proceed to execute and analyze three distinct experiments. Each experiment is designed with a specific objective in mind, and the following sections will provide a detailed account of the experiments, their results, and the associated analyses.

### 4.1. Experimental Design

The experiments were performed in a Python 3.10 environment using signal data from the RML2016.10a dataset [30]. Signals from three modulation types (QAM16, BPSK, and QPSK) were linearly mixed, with the additive white Gaussian noise already present in the dataset. To facilitate experimental validation, the center frequency of QAM16-modulated, BPSK-modulated, and QPSK-modulated signals were adjusted from 0 Hz to 5 Hz, 30 Hz, and 20 Hz, respectively, as all signals in the dataset initially had a center frequency of 0 Hz. Simulation experiments were conducted to compare the separation and denoising performance of three signal types mixed at 10 dB; furthermore, the proposed algorithm was assessed against the AVMD-based SCBSS algorithm and the GWO-VMD-based SCBSS algorithm. Furthermore, the trend of separation performance was examined across varying SNRs for the SCBSS algorithms, and experiments compared separation performance across different algorithms under the same modulation types. The aim was to validate the proposed algorithm’s accuracy, precision, and adaptability.

### 4.2. Analysis of Separation and Denoising Performance of the Proposed Algorithm

The investigation involved the random selection of 384 consecutive points from the three signal types, as described in the aforementioned design schemes, with a signal-to-noise ratio (SNR) of 10 dB. These signals were subjected to linear mixing based on preset weights. Experimental analysis revealed that the weighting coefficients had minimal impact on the separation effectiveness, resulting in proportional mixing of the signal types. The original and mixed signals are illustrated in Figure 3.

Figure 3 illustrates the modulation schemes of the signals: QAM16 for signal 1, BPSK for signal 2, and QPSK for signal 3, all at a 10 dB SNR. Following equal-weight mixing, they formed the combined signal. Through the application of the IPSO-optimized VMD algorithm with 5 iterations, the fitness function reached a minimum value of −1.028 $\times \;10^{19}$. This corresponded to the optimal parameter set for the VMD algorithm as (6, 2289).

Table 1 presents a comparative analysis of the IPSO algorithm against the standard PSO algorithm. It is observed that at the 100th iteration, the standard PSO algorithm fell into local optima on four occasions. In contrast, the IPSO algorithm did not experience any instances of local optimal trapping. This demonstrates the superior performance of the improved PSO algorithm in addressing the single-channel blind source separation issue discussed within this study.

Upon determining the optimal combination of (*k*, α) as (6, 2289), Figure 4a,b illustrate the time-domain and frequency-domain plots, respectively, of the VMD decomposition of the mixed signal. The initial three components display noticeable central frequencies, whereas the fourth component lacks a clear central frequency, suggesting potential noise. Utilizing the SE introduced in Section 3.3 to sift through valid components revealed that the SE values of the first three components fell below the predetermined threshold, aligning with the characteristics of valid modal components.

As a result, the signal’s reconstruction using these components and its comparison with the original mixed signal are illustrated in Figure 5a,b.

Figure 5a,b illustrate a close resemblance between the reconstructed signal and the original mixed signal in the time domain. In Figure 5b, the signal reconstructed from the first three components after VMD closely approximates the original mixed signal, with a similarity coefficient of 0.98288.

Figure 6a–c depict the comparison of similarity between the separated signals and their respective source signals, yielding correlation coefficients of 0.95589, 0.93715, and 0.95376, respectively.

The level of resemblance observed between the separated signal components and their respective source signals suggests that the proposed algorithm can effectively isolate the individual source signals.

The SNR of the separated independent signals was evaluated within the SNR range of the source signal, spanning from –20 dB to 10 dB. The results of this evaluation are presented in Table 2, which provides a detailed illustration of the SNR for the separated independent components across different modulation schemes and various SNR levels of the source signals.

The comparative trend of the signal-to-noise ratios (SNRs) between the independent components and the source signals is illustrated in Figure 7.

Figure 7 depicts the variation tendency of the signal-to-noise ratio (SNR) of the independent components in relation to the SNR of the source signal. Evidently, at low SNR levels, across all three modulation schemes (QAM16, BPSK, and QPSK), the SNR of the independent signals isolated by our proposed algorithm exhibits a remarkable enhancement compared to that of the source signals. Remarkably, the maximum increment can attain 24.66 dB, which convincingly attests to the potent noise suppression capacity of the algorithm. Particularly for the QAM16 modulated signals, this improvement is especially conspicuous. This could be ascribed to the algorithm’s superior adaptability to the intricate structure and characteristics of the QAM16 signals, thereby effectively eliminating the noise components and elevating the signal quality.

Nevertheless, when the SNR of the source signal surpasses 8 dB, the noise reduction effect gradually diminishes under all modulation schemes. This occurrence implies that in a high-SNR environment, the superiority of the algorithm in noise suppression is not as salient as it is at low SNR. The probable cause is that in a high-SNR scenario, the signal quality is relatively high, and the impact of noise on the signal is relatively minor, resulting in a restricted scope for the algorithm to further enhance the SNR.

### 4.3. Comparison Analysis of the Proposed Algorithm with Three Different Separation Algorithms at the Same SNR

This paper proceeds to compare the proposed algorithm with the AVMD-based SCBSS algorithm [6], SCBSS algorithm with parameter set values determined empirically [31], and GWO-VMD-based SCBSS algorithm by analyzing the average correlation coefficient between the separated independent signals and the source signals across different SNRs ranging from –15 dB to 10 dB, as shown in Figure 8.

Simultaneously, the separation performance of the four algorithms was compared using the unit average correlation coefficient (average correlation coefficient per decibel) and the signal-to-noise ratio improvement as evaluation metrics, as shown in Table 3.

As depicted in Figure 8, the proposed algorithm attains an average correlation coefficient exceeding 0.75 with the source signals at an SNR of –15 dB. The AVMD-based SCBSS algorithm, which adaptively determines the value of k, failed to achieve a correlation coefficient exceeding 0.2. The SCBSS algorithm with empirically determined parameter sets exhibited the lowest correlation coefficients within the SNR ranging from 0 dB to 10 dB compared to the other three methods. As depicted in Table 3, The GWO-VMD-based SCBSS algorithm, which adaptively determines the parameter set, demonstrated an average correlation coefficient marginally lower than that of the algorithm presented herein, yet its outcomes were less stable than those of the proposed methodology. From the calculation time of the three algorithms, ours algorithm is superior to the other three methods in convergence speed. Moreover, across the entire range of SNR fluctuations, the proposed algorithm consistently outperforms the other two methods of empirically determining parameters and adaptively determining k values in terms of separation effectiveness, with a minimum correlation coefficient improvement of 15.7%, and the separation results are more stable than the same method of adaptively determining parameter groups.

### 4.4. Comparison of Separation and Denoising Effects of Different Separation Algorithms Under the Same Modulation Scheme

This study also evaluates the effectiveness of the proposed algorithm against single-channel BSS algorithms utilizing adaptive Morlet wavelets, multi-scale Morlet wavelets, and Morlet wavelets [32]. The comparative outcomes are illustrated in Figure 9a,b.

Figure 9a depicts the correlation coefficient trend relative to the SNR for BPSK-modulated signals, while Figure 9b depicts the same for QPSK modulation. Both figures illustrate that as the SNR of the source signal increases, the correlation coefficients of the independent components separated by the proposed algorithm and the three algorithms cited in [32] also increase in relation to the source signals. The algorithms mentioned in [32] exhibit better reconstruction performance for QPSK-modulated signals, maintaining a correlation coefficient exceeding 0.65 even under low-SNR conditions. Conversely, for BPSK-modulated signals, the maximum achievable correlation coefficient is only 0.85 at high SNRs. The proposed algorithm consistently upholds a correlation coefficient surpassing 0.9 under different modulation schemes and varying SNR conditions. This highlights the efficacy of resolving the BSS issue for single-channel mixed signals.

### 4.5. Summary

This study presents an IPSO-VMD algorithm, which adaptively determines the optimal combination of mode component number (k) and quadratic penalty factor (α), mitigating mode mixing and endpoint effects in decomposition. Additionally, the IPSO algorithm decreases the likelihood of the PSO algorithm getting trapped in local optima. Using the IFastICA algorithm, it separates independent signals that are highly correlated and phase-coherent with source signals from virtual multi-channel data comprising effective mode components and observed signals. Moreover, the IPSO-VMD algorithm achieves denoising, enhancing the maximum SNR by up to 24.66 dB, showcasing its utility in solving the BSS problem for single-channel signals. Compared to the existing AVMD-based SCBSS and the SCBSS algorithms with parameter set values determined empirically, the IPSO-VMD algorithm’s separation effectiveness improves by at least 15.7%. Additionally, compared to SCBSS algorithms based on adaptive Morlet wavelets, multi-scale Morlet wavelets, and Morlet wavelets, our proposed algorithm demonstrates robust separation performance for signals with modulation schemes such as BPSK and QPSK; furthermore, the algorithm maintains correlation coefficients above 0.9 and superior separation and denoising effects under various modulation schemes and SNRs.

## 5. Conclusions

In this study, we propose an adaptive SCBSS algorithm that integrates IPSO and VMD for single-antenna or single-device Edge AI communication systems. The algorithm was validated using the RML2016.10a dataset, demonstrating a remarkable correlation coefficient exceeding 0.9 between the separated components and the source signals, highlighting its exceptional separation performance and ability to preserve signal integrity. When compared to existing methods, the proposed algorithm achieves a separation improvement of approximately 15.7%, which is critical in practical applications such as wireless sensor networks and military communication. However, the algorithm exhibits certain limitations. Its performance may degrade in highly non-stationary environments and under extremely low-SNR conditions. Future work will focus on incorporating deep learning techniques to further enhance the algorithm’s adaptability, accuracy, and robustness. Additionally, efforts will be directed towards reducing computational complexity through parallel computing and algorithm optimization, ensuring real-time processing capabilities for Edge AI systems.

## Figures and Tables

**Figure 1 sensors-25-01107-f001:**
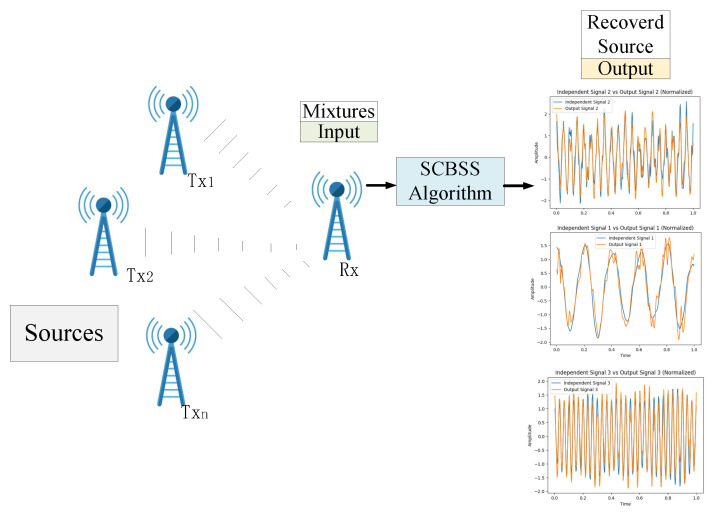
The process of single-channel blind source separation (SCBSS) within a wireless communication system.

**Figure 2 sensors-25-01107-f002:**
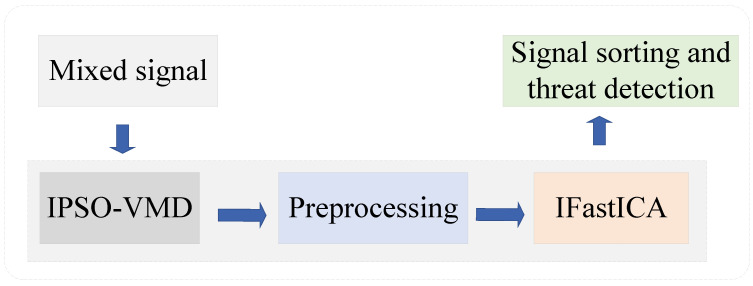
Flowchart of single-channel blind source separation (SCBSS).

**Figure 3 sensors-25-01107-f003:**
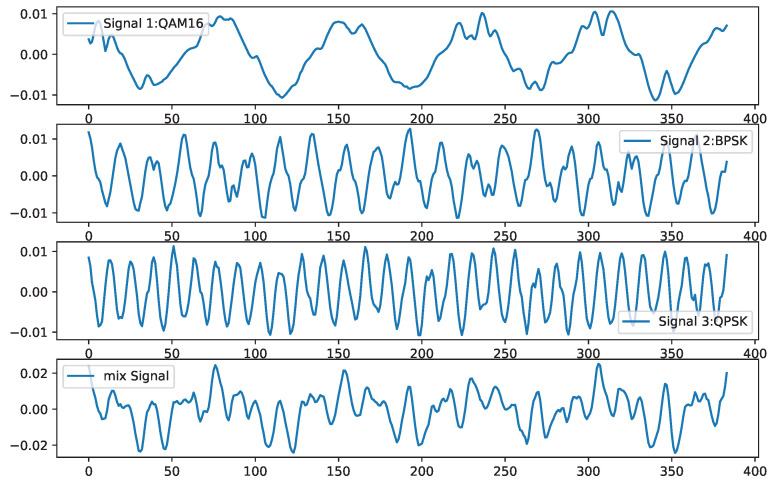
Original and mixed signals.

**Figure 4 sensors-25-01107-f004:**
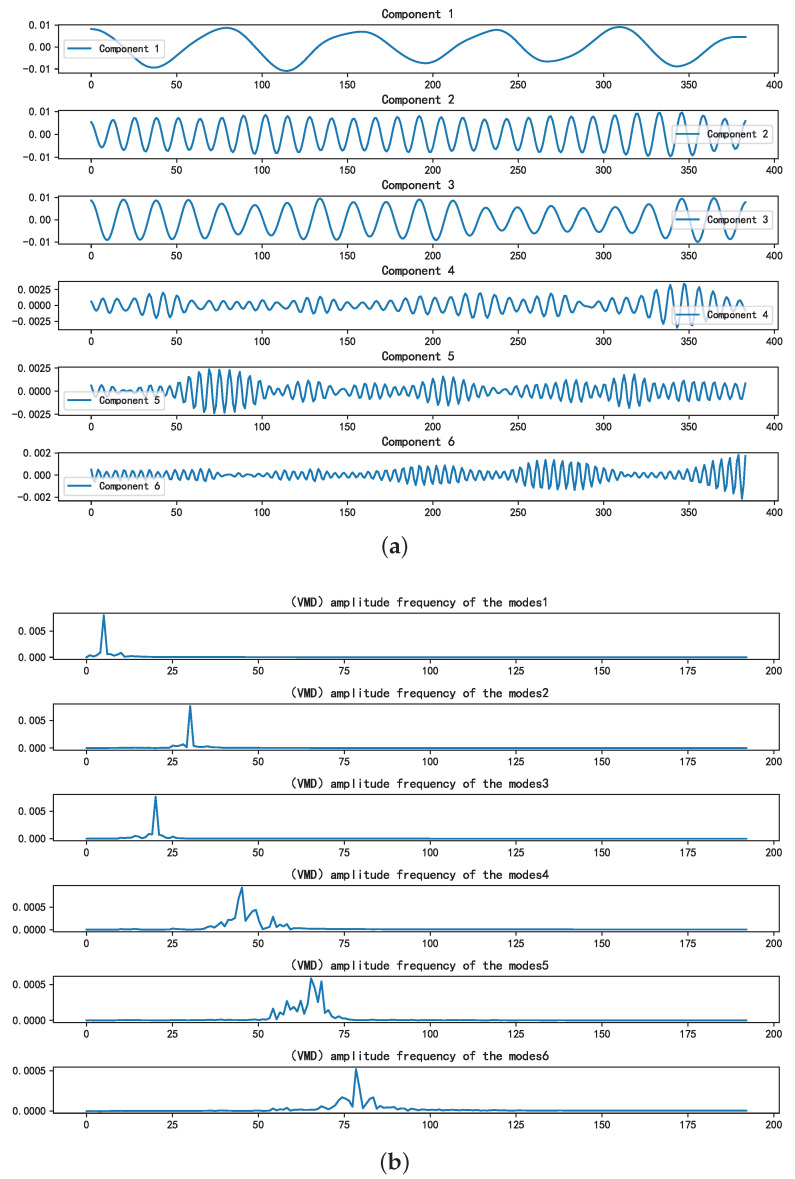
The time-frequency domain representions of the IMFs. (**a**): Time-domain representation of the intrinsic mode functions. (**b**): Frequency-domain representation of the IMFs.

**Figure 5 sensors-25-01107-f005:**
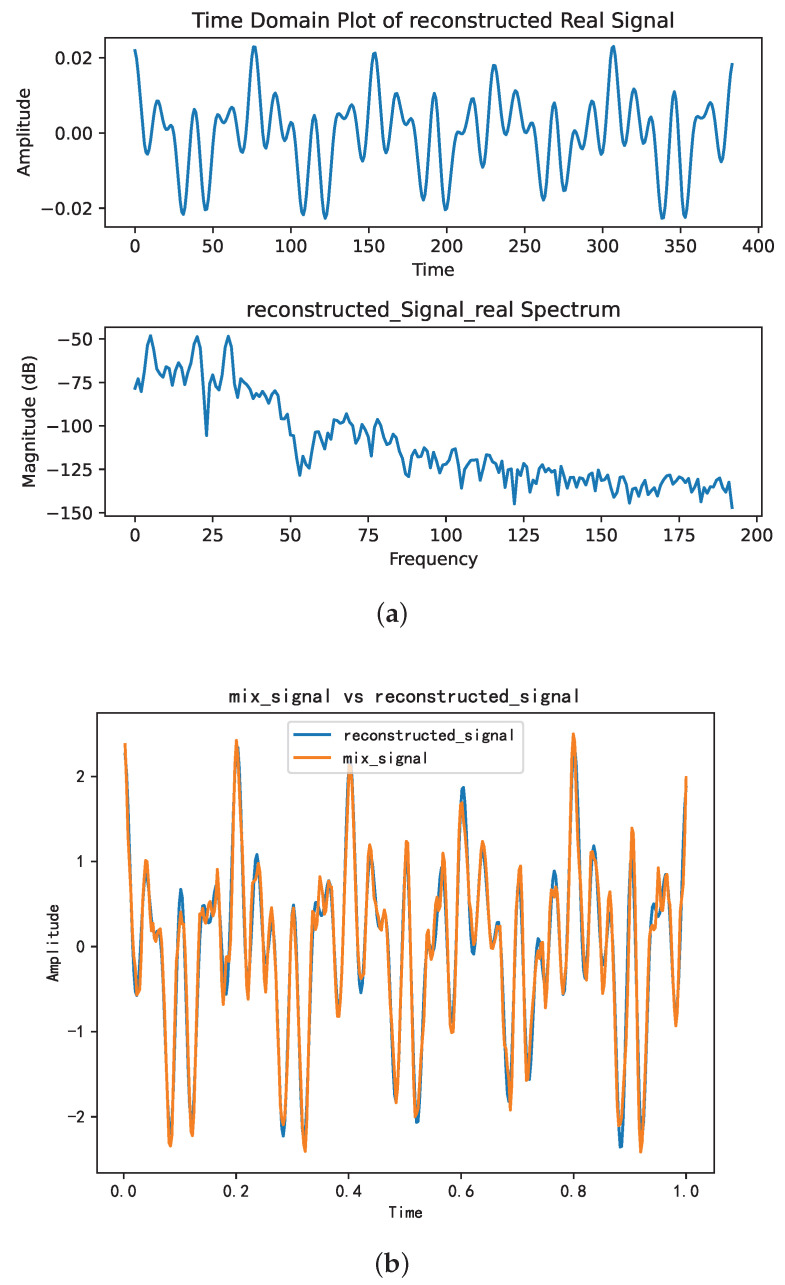
The time-frequency domain of the reconstructed signal. (**a**) Time-frequency representation of the reconstructed signal. (**b**) Comparative analysis of the reconstructed signal versus the mixture signal.

**Figure 6 sensors-25-01107-f006:**
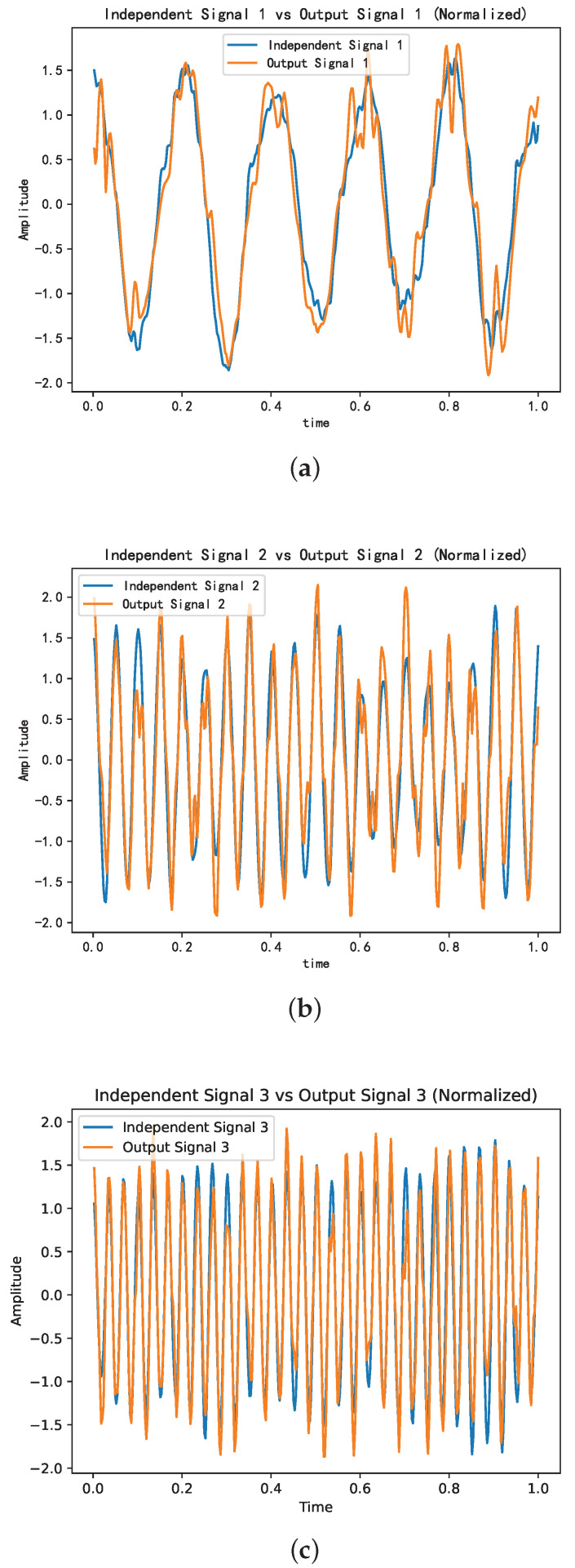
Comparative analysis of the independent signals versus the source signals. (**a**) Comparative analysis of independent signal 1 versus source signal 1. (**b**) Comparative analysis of independent signal 2 versus source signal 2. (**c**) Comparative analysis of independent signal 3 versus source signal 3.

**Figure 7 sensors-25-01107-f007:**
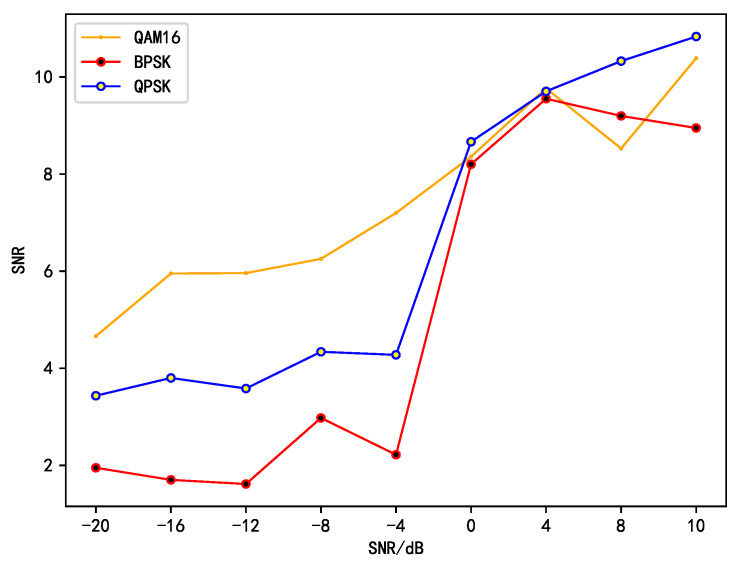
The variation trend of independent component signal-to-noise ratio (SNR) with source signal SNR.

**Figure 8 sensors-25-01107-f008:**
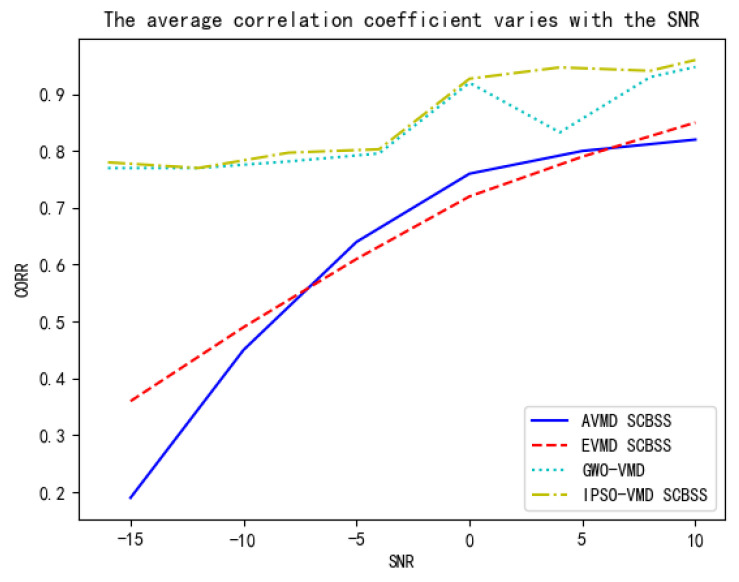
Trend of average correlation coefficient with SNR variations.

**Figure 9 sensors-25-01107-f009:**
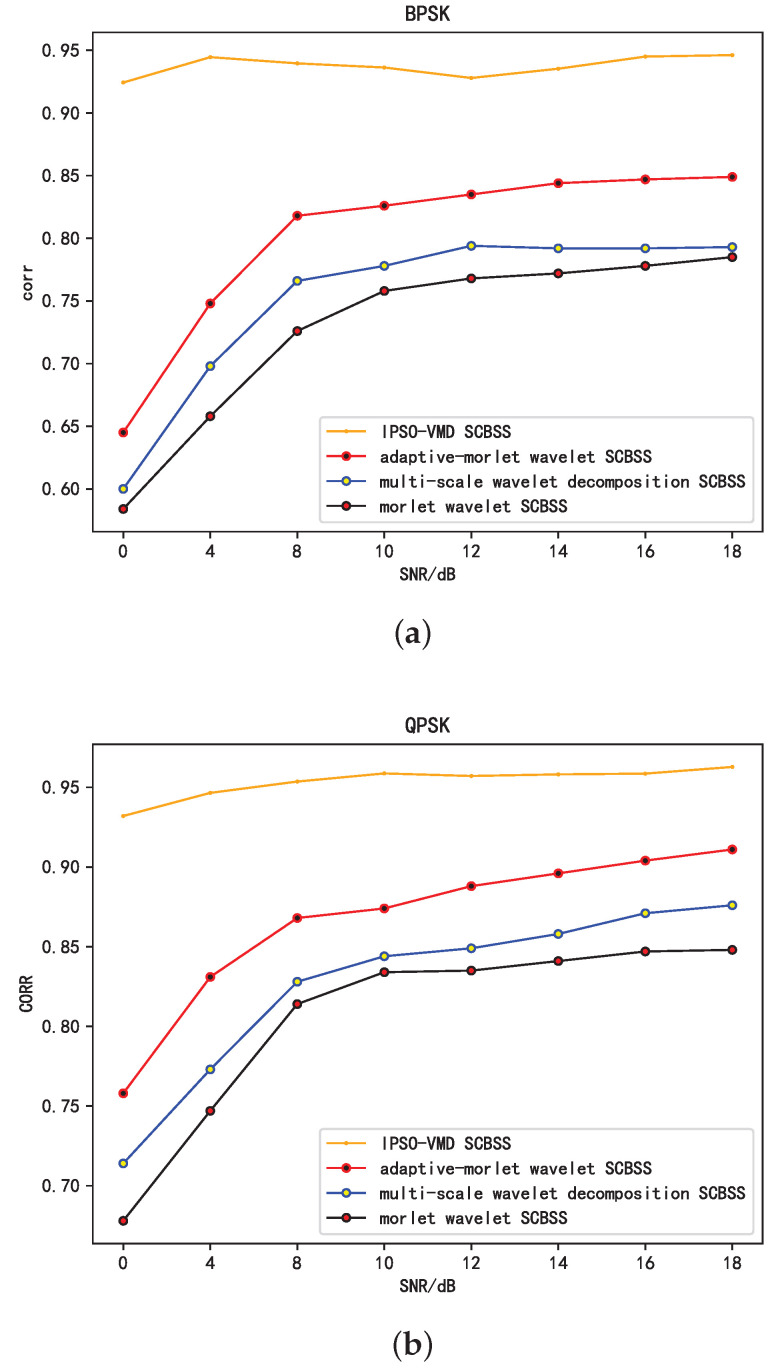
Comparison of the separation effects of different methods under the same modulation mode. (**a**) Trend of average correlation coefficient with SNR variations (BPSK). (**b**) Trend of average correlation coefficient with SNR variations (QPSK).

**Table 1 sensors-25-01107-t001:** Comparative analysis of the IPSO algorithm against the standard PSO algorithm.

Algorithm	Iterations	Local Optima Hits	Fitness Function Optimal Value
IPSO Algorithm	100	0	−1.0279 $\times \;10^{19}$
PSO Algorithm	100	4	−1.0279 $\times \;10^{19}$

**Table 2 sensors-25-01107-t002:** Variation of independent component signal-to-noise ratio (SNR) with source signal SNR.

SNR (dB)	−20	−16	−12	−8	−4	0	4	8	10
Method									
QAM16	4.66	5.95	5.96	6.27	7.20	8.36	9.76	8.53	10.39
BPSK	3.43	3.80	3.58	3.72	4.28	8.67	9.70	10.33	10.83
QPSK	1.94	1.70	1.61	1.41	2.22	8.20	9.55	9.20	8.95

**Table 3 sensors-25-01107-t003:** Unit mean correlation coefficient for different algorithms.

Algorithm	Unit Mean Correlation Coefficient	Computing Time
AVMD SCBSS [6]	0.0350	2.011
EVMD SCBSS [31]	0.0411	1.929
GWO-VMD SCBSS	0.0482	1.033
**Ours (IPSO-VMD SCBSS)**	**0.0497**	**0.864**

## Data Availability

The data are not publicly available due to other purposes.

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
