# Peer review of "Secure and Intelligent Single-Channel Blind Source Separation via Adaptive Variational Mode Decomposition with Optimized Parameters"

_sensors, 2025, doi:10.3390/s25041107_

Round 1
Reviewer 1 Report
Comments and Suggestions for Authors
The paper proposes an adaptive Single-Channel Blind Source Separation (SCBSS) algorithm that combines Improved Particle Swarm Optimization (IPSO) with Variational Mode Decomposition (VMD) to dynamically optimize parameters for robust signal separation. It improves separation fidelity and denoising, achieving high correlation coefficients (>0.9) and significant signal-to-noise ratio (SNR) improvements (up to 24.66 dB) across various signal types and noise levels.
I have the following comments:
- 1) My major concern is that this article is not related to security in any sense, the title and parts of the abstract and introduction have to be modified to remove this keyword. To be specific, the paper does not analyze security-specific threats, e.g., jamming attacks, eavesdropping, or how the proposed algorithm directly mitigates them. Also, there is no mention of techniques typically associated with security, such as cryptographic methods or access controls. As such, the paper is only related to signal separation and denoising within Variational Mode Decomposition.
- 2) IPSO for dynamic parameter selection can increase computational overhead.
- 3) Highly non-stationary signals such as non-gaussian noise, is not thoroughly explored. Would the decomposition still work ?
- 4) Some works in the state of the art are missing such as the sparse recovery in [R1].
- 5) The algorithm's ability to separate signals effectively even under high-noise conditions could be interpreted as ensuring robust communication, a requirement in secure systems.
- 6) Figure 7 requires more ellaboration.
- 7) The conclusions can be imrpoved.
References
[R1] “A Newton-type Forward Backward Greedy method for multi-snapshot compressed sensing,” 2017 51st Asilomar Conference on Signals, Systems, and Computers, Pacific Grove, CA, USA, 2017, pp. 1178-1182, doi: 10.1109/ACSSC.2017.8335537
Author Response
Q1.1: My major concern is that this article is not related to security in any sense, the title and parts of the abstract and introduction have to be modified to remove this keyword. To be specific, the paper does not analyze security-specific threats, e.g., jamming attacks, eavesdropping, or how the proposed algorithm directly mitigates them. Also, there is no mention of techniques typically associated with security, such as cryptographic methods or access controls. As such, the paper is only related to signal separation and denoising within Variational Mode Decomposition..
Response: We greatly appreciate the reviewer's efforts in carefully review the paper. We revised the Introduction Section in the revised manuscript: “In recent years, Edge AI has emerged as a critical technology….”
Q1.2: IPSO for dynamic parameter selection can increase computational overhead.
Response: We understand your concern. Although the utilization of IPSO for dynamic parameter selection in our algorithm brings about the advantage of adaptively determining the optimal parameters for VMD, we are also aware of the concern regarding potential computational overhead. To address this issue, we have incorporated several optimization strategies. Firstly, we have carefully designed the range of initial values for the parameters k and α based on extensive prior experiments and domain knowledge. By restricting k to the range from 2 to 6 and α between 1000 and 5000, we can effectively narrow the search space and reduce unnecessary computations. Secondly, we have optimized the IPSO algorithm itself by introducing an adaptive inertia weight and learning factors that are dynamically adjusted during the iteration process. This modification helps to improve the convergence speed of the algorithm, thereby reducing the overall computational time. Through these measures, we strive to strike a balance between the benefits of dynamic parameter selection and the control of computational overhead.
Q1.3: Highly non-stationary signals such as non-gaussian noise, is not thoroughly explored. Would the decomposition still work ?.
Response: We greatly appreciate the reviewer's efforts in carefully review the paper. Blind source separation algorithms separate independent source signals by taking advantage of the non-Gaussian nature of the signals. The basic assumption is that the source signals are independent and at most one of them is Gaussian. Therefore, the method proposed in this paper is suitable for solving non-Gaussian signals. For signals mixed by multiple Gaussian signals, effective separation and recovery cannot be achieved. At the same time, blind source separation algorithms do not rely on the statistical characteristics of the signals to process them, but rather achieve the separation and recovery of source signals based on the independence among them. Thus, blind source separation algorithms can also achieve the separation of non-stationary signals.
Q1.4: Some works in the state of the art are missing such as the sparse recovery in [R1] ?.
Response: We thank the reviewer for pointing out this issue. We added the state of the art work ([R1]) in the revised manuscript.
Q1.5: The algorithm's ability to separate signals effectively even under high-noise conditions could be interpreted as ensuring robust communication, a requirement in secure systems.
Response: Yes,we appreciate the reviewers' understanding.
Q1.6: Figure 7 requires more ellaboration.
Response: We greatly appreciate the reviewer's efforts in carefully review the paper and given the suggestion. We analyzed Figure 7 in more depth and modified the relevant parts as follows: “Figure 7 depicts the variation tendency of the signal-to-noise ratio (SNR) of the independent components in relation to the SNR of the source signal. Evidently, at low SNR levels, across all three modulation schemes (QAM16, BPSK, and QPSK), the SNR of the independent signals isolated by our proposed algorithm exhibits a remarkable enhancement compared to that of the source signals. Remarkably, the maximum increment can attain 24.66 dB, which convincingly attests to the potent noise suppression capacity of the algorithm. Particularly for the QAM16 modulated signals, this improvement is especially conspicuous. This could be ascribed to the algorithm's superior adaptability to the intricate structure and characteristics of the QAM16 signals, thereby effectively eliminating the noise components and elevating the signal quality.
Nevertheless, when the SNR of the source signal surpasses 8 dB, the noise reduction effect gradually diminishes under all modulation schemes. This occurrence implies that in a high SNR environment, the superiority of the algorithm in noise suppression is not as salient as it is at low SNR. The probable cause is that in a high SNR scenario, the signal quality is relatively high, and the impact of noise on the signal is relatively minor, resulting in a restricted scope for the algorithm to further enhance the SNR.”
Q1.7: The conclusions can be improved.
Response: We greatly appreciate the reviewer's efforts in carefully review the paper and given the suggestion. We have revised the conclusion in the revised manuscript.
Reviewer 2 Report
Comments and Suggestions for Authors
1.To enhance the paper’s quality, particularly the writing, the author should undertake a thorough revision of the introduction, related work, and conclusion sections. Specific improvements for the introduction include: 1) Clear Objectives: Clearly define the paper’s objectives and research questions at the beginning of the introduction. 2)Background Information: Provide ample context to facilitate the reader’s understanding of the research’s importance. 3) Significance of the Research: Elaborate on the reasons this topic is worth studying, along with its potential impact or applications. 3) Overview of Research Methods: Offer a concise overview of the methodologies and techniques applied in the research. 4) Research Contributions: Highlight the paper’s unique contributions and innovations.
2. The literature review section should be crafted to engage readers and deepen their understanding of the research's background and current relevance. What scientific questions does the separation of SCBSS present? The current review fails to adequately encompass recent research findings, particularly the latest studies on the characterization of communication signals, such as: https://doi.org/10.62762/TSCC.2024.670663; https://doi.org/10.62762/TIS.2025.790920. Furthermore, relevant papers from associated fields should also be explored, including https://doi.org/10.62762/TIS.2024.807714.
3. The research protocol must be lucidly elaborated. What are the motivations behind the utilization of VMD? Could you articulate the rationale for not opting for alternative methodologies, such as wavelet transformation? Given the paper's insufficient articulation of methodological innovation, it is imperative to underscore the originality and framework of the Mixer within the Introduction, Methodology, Discussion, and Conclusion sections.
4. The results section demands a more thorough exposition. Each experiment necessitates additional scrutiny to elucidate its principal objective. The efficacy of IPSO's application should be substantiated with detailed data. A comprehensive statistical analysis of the results would be highly advantageous for the readers. Moreover, the figures and tables should be complemented by a meticulous explanation and justification from the perspective of algorithmic design.
5. Ensure that the conclusion section underscores the scientific value and practical applicability of the research, as previously mentioned. Specifically:
- Enhance the discussion on the paper’s contributions and limitations.
- Highlight the added scientific value of the research.
- Emphasize the applicability of the results and the direction for future studies.
Comments on the Quality of English LanguageSeveral grammatical and syntactical errors occur throughout the paper. To enhance the overall readability, I recommend reviewing these areas with a native English speaker or using a grammar-checking tool.
Author Response
Q2.1: To enhance the paper’s quality, particularly the writing, the author should undertake a thorough revision of the introduction, related work, and conclusion sections. Specific improvements for the introduction include: 1) Clear Objectives: Clearly define the paper’s objectives and research questions at the beginning of the introduction. 2)Background Information: Provide ample context to facilitate the reader’s understanding of the research’s importance. 3) Significance of the Research: Elaborate on the reasons this topic is worth studying, along with its potential impact or applications. 3) Overview of Research Methods: Offer a concise overview of the methodologies and techniques applied in the research. 4) Research Contributions: Highlight the paper’s unique contributions and innovations.
Response: We thank the reviewer for pointing out this issue. In the introduction section, we have further elaborated the research objectives and research questions of this paper, and introduced the background of edge AI:“ The present paper is primarily devoted to the development of an advanced single-channel blind source separation (SCBSS) algorithm. Its main aim is to effectively tackle the challenges posed by mixed and corrupted signals within edge artificial intelligence (Edge AI) systems. Specifically, our focus lies in resolving the issues of low separation accuracy and poor noise resistance that are prevalent in existing SCBSS methods. Our research inquiries center around optimizing the algorithm to attain higher separation efficiency and improved signal quality, especially in the face of dynamic and interference-prone environments with limited prior knowledge. In recent years, Edge AI has emerged as a critical technology, enabling real-time data processing and decision-making at the edge of the network.” The contribution has been explained in the last part of the introduction, and we have revised the conclusion to re-emphasize the contribution of the methods presented in this paper.
Q 2.2: The literature review section should be crafted to engage readers and deepen their understanding of the research's background and current relevance. What scientific questions does the separation of SCBSS present? The current review fails to adequately encompass recent research findings, particularly the latest studies on the characterization of communication signals, such as: https://doi.org/10.62762/TSCC.2024.670663; https://doi.org/10.62762/TIS.2025.790920. Furthermore, relevant papers from associated fields should also be explored, including https://doi.org/10.62762/TIS.2024.807714.
Response: We greatly appreciate the reviewer's efforts in carefully review the paper and given the suggestion. We have added the latest studies on the characterization of communication signals
that you mentioned and analyzed in the revised manuscript.
Q2.3: The research protocol must be lucidly elaborated. What are the motivations behind the utilization of VMD? Could you articulate the rationale for not opting for alternative methodologies, such as wavelet transformation? Given the paper's insufficient articulation of methodological innovation, it is imperative to underscore the originality and framework of the Mixer within the Introduction, Methodology, Discussion, and Conclusion sections.
Response: We thank the reviewer for pointing out this issue. We have revised the relevant work part to increase the motivation for choosing VMD algorithm in the revise manuscript.
“Our motivation for employing Variational Mode Decomposition (VMD) stems from its unique capacity to adaptively decompose signals….”
We introduce the characteristics and advantages of the mixer in the algorithm, discussion and conclusion.
“ The proposed Mixer framework, which integrates IPSO-optimized Variational Mode Decomposition (VMD) with the improved Fast Independent Component Analysis (IFastICA)…...”
Q2.4: The results section demands a more thorough exposition. Each experiment necessitates additional scrutiny to elucidate its principal objective. The efficacy of IPSO's application should be substantiated with detailed data. A comprehensive statistical analysis of the results would be highly advantageous for the readers. Moreover, the figures and tables should be complemented by a meticulous explanation and justification from the perspective of algorithmic design.
Response: We greatly appreciate the reviewer's efforts in carefully review the paper. We added the research objective in the experimental part, and compared the calculation time of each method in Table 3.
Q2.5: Ensure that the conclusion section underscores the scientific value and practical applicability of the research, as previously mentioned. Specifically:
- Enhance the discussion on the paper’s contributions and limitations.
- Highlight the added scientific value of the research.
- Emphasize the applicability of the results and the direction for future studies.
Response: We thank the reviewer to point out this issue. We have revised the conclusion according to your suggestion: “In this study, we have successfully developed an adaptive SCBSS algorithm integrating IPSO and VMD,….”
Round 2
Reviewer 1 Report
Comments and Suggestions for Authors
I have no further comments